# Ultrasonic Production of Chitosan Nanoparticles and Their Application Against *Colletotrichum gloeosporioides* Present in the Ataulfo Mango

**DOI:** 10.3390/polym16213058

**Published:** 2024-10-30

**Authors:** Ivana Solis Vizcaino, Efraín Rubio Rosas, Eva Águila Almanza, Marco Marín Castro, Heriberto Hernández Cocoletzi

**Affiliations:** 1Facultad de Ingeniería Química, Benemérita Universidad Autónoma de Puebla, Av. San Claudio y 18 sur S/N Edificio FIQ7 CU San Manuel, Puebla 72570, Mexico; sv224570055@alm.buap.mx; 2Dirección de Innovación y Transferencia de Conocimiento, Benemérita Universidad Autónoma de Puebla, Prol. 24 sur S/N CU San Manuel, Puebla 72570, Mexico; efrain.rubio@correo.buap.mx; 3Centro de Investigación en Ciencias Agrícolas, Benemérita Universidad Autónoma de Puebla, ICUAP, 14 sur 6301, San Manuel, Puebla 72570, Mexico; marco.marin@correo.buap.mx

**Keywords:** Ataulfo mango, chitosan, chitosan nanoparticles, ultrasound, anthracnose

## Abstract

In Mexico, the Ataulfo mango crop faces significant challenges due to anthracnose, a disease caused by the fungus *Colletotrichum gloeosporioides*. The need to use eco-friendly fungicides is crucial to avoid the use of harmful synthetic chemicals. This study aimed to prepare chitosan nanoparticles through a simple and effective ultrasound-assisted top-down method, with high antifungal efficiency. The nanoparticles were prepared from chitosan (DD = 85%, MW = 553 kDa) and Tween 20 under constant sonication. The formation of the nanoparticles was initially confirmed by Fourier-transform infrared (FTIR) spectroscopy; and their physicochemical properties were subsequently characterized using scanning electron microscopy (SEM) and atomic force microscopy (AFM). The antifungal potential of the chitosan nanoparticles against the phytopathogen *Colletotrichum gloeosporioides* was evaluated with isolated fungi obtained directly from mango tissues showing anthracnose symptoms in the state of Guerrero, Mexico. The fungus was identified through SEM imaging, showing a regular and smooth conidial layer, with cylindrical shape (r = 2 µm, h = 10 µm). In vitro tests were conducted with three different concentrations of chitosan nanoparticles to assess their inhibitory effects. After seven days of incubation, a maximum inhibition rate of 97% was observed with the 0.5% nanoparticle solution, corresponding to a fungal growth rate of 0.008 cm/h. At this time, the control mycelial growth was 7 cm, while the treated sample reached a radius of 0.55 mm. These results demonstrated the antifungal effect of the nanoparticles on the membrane and cell wall of the fungus, suggesting that their composition could induce a resistance response. The inhibitory effect was also influenced by the particle size (30 nm), as the small size facilitated penetration into fungal cells. Consequently, the parent compound could be formulated and applied as a natural antifungal agent in nanoparticle form to enhance its activity. The method described in this study offers a viable alternative for the preparation of chitosan nanoparticles, by avoiding the use of toxic reagents.

## 1. Introduction

The Ataulfo mango (*Mangifera caesia Jack ex Wall*) is one of the most sought-after seasonal fruits due to its flavor and properties [1,2]. In 2023, Mexico was among the six main producing countries of this species, occupying the fifth place [3]. The state of Guerrero is the main producer, contributing 22% of the total production, with an estimated 364 thousand tons. However, like all fruits, this one is susceptible to diseases, either in the plant or in the fruit. It is usually affected by anthracnose, produced by the fungus *Colletotrichum gloeosporioides* [4]; this is considered the most important disease in the pre- and post-harvest stages. The Costa Grande experiences the most significant damage (from December to April); the losses recorded are up to 60% of the production [5]. A variety of chemical fungicides have been used to combat anthracnose; however, up to 98% usually do not reach their target species, causing environmental pollution and severe damage to human health [6]. This encourages the search for and study of new alternatives that benefit sustainable agriculture and reduce the use of harmful materials.

In this sense, the physicochemical properties, the activation of new reactive groups and a greater surface area allow the nanoparticles to be more effective against pathogens [7]. Chitosan-based nanoparticles (ChNPs) are preferentially used worldwide for various applications due to their biodegradability, high permeability, non-toxicity to humans and cost-effectiveness [8].

ChNPs have been used in agriculture as an antifungal agent, because they can control and even eradicate different pathogens that attack plants. ChNPs have been applied individually or in combination with other natural fungicides [9]. Through in vitro tests, Wook et al. [10] found the inhibitory effect of chitosan nanoparticles against *Colletotrichum gelosporidies*, *Phytophthora capsici*, *Sclerotinia sclerotiorum*, *Fusarium oxysporum*, and *Gibberella fujikuori* commonly found in tomatoes. In vitro studies have shown high efficiency in the inhibition of *Colletotrichum fragariae* isolated from strawberries, when using ChNPs in combination with blueberry extracts [11]; likewise, the inhibition of the mycelium of *Botrytis cinerea* present in different plants has been reported when using ChNPs in conjunction with thymol [12]. Gold particles have also been incorporated into ChNPs with the purpose of protecting tomato crops affected by *Fusarium oxysporum* [13]; to achieve the inhibitory effect, it was necessary to apply 5 mL of solution.

While chitosan has been shown to have wide antifungal applications [14], their insolubility of bulk in aqueous medium limits their wide spectrum of applications as an antifungal agent, giving rise to ChNPs, which have emerged as an effective strategy due to their antifungal properties, low production cost and zero toxicity [15]. There are various methods used for the synthesis of chitosan nanoparticles, the most common being ionic gelation, microemulsion, emulsion-based solvent evaporation and emulsion solvent diffusion [16]; however, these techniques have high disadvantages, such as difficulty in obtaining NPs of uniform size, the use of harmful chemicals, significant shear forces produced during the formation of CNPs, in addition to a substantial reduction in hydrophilicity; so, in recent years efforts have been made to develop environmentally friendly techniques [17]. The implementation of ultrasound in the deproteinization stage has allowed for chitosan to be obtained with different molecular weights and degrees of deacetylation, as well as a reduction in the particle size [18]; this has allowed for savings in reagents and energy. Particle size reduction has also been attempted by incorporating surfactants such as Tween 80 [19].

Chitosan, a biocompatible polymer derived from the deacetylation of chitin, has a cellulose-like carbohydrate foundation structure with two types of alternating repeating units, glucosamine units and N-acetyl glucosamine, linked by a 1–4-glycosidic linkage, and is a whitish and inelastic polysaccharide [20] that exhibits potent antifungal activity against various plant pathogens [21]. Its mechanism of action involves disrupting fungal plasma membranes, increasing intracellular reactive oxygen species, and altering gene expression [22,23]. Chitosan nanoparticles demonstrate enhanced antifungal effects compared to regular chitosan, completely inhibiting mycelial growth at lower concentrations [21]. The antimicrobial efficacy of chitosan is attributed to its chelating ability and positively charged amino groups, which interact with negatively charged microbial surfaces [24]. However, the antifungal activity can be influenced by factors related to the chitosan molecule, the target fungus, and environmental conditions [23]. Interestingly, chitosan also modulates plant defense responses and affects auxin accumulation in plant roots [22]. These findings highlight the potential of chitosan and its nanoparticles for agricultural applications.

To our knowledge, no strategies have been implemented to avoid the use of chemicals to control *Colletotrichum gloeosporioides*, which has harmful effects on Ataulfo mango. In this work, the obtaining of chitosan nanoparticles using ultrasound and Tween 20 is reported; through in vitro studies, its antifungal action against the fungus *Colletotrichum gloeosporioides* is reported.

## 2. Materials and Methods

### 2.1. Materials

Chitosan polymer with an average molecular weight (MW) of 553 kDa and a degree of deacetylation (DD) of 85% was obtained in a previous work [18]. Glacial acetic acid (99.7%), glycerol (99%), anhydrous sodium acetate (≥99%) and Tween 20 were supplied by Sigma Aldrich Chemical Co. (St. Louis, MO, USA)

### 2.2. Preparation of Chitosan Nanoparticles

Chitosan nanoparticles (ChNPs) were obtained according to the methodology proposed by [25] with some modifications. The 0.25%, 0.5%, and 1% (*w*/*v*) chitosan solutions were prepared in a 2% glacial acetic acid solution/1% (*v*/*v*) Tween 20 solution. To 200 mL of each chitosan solution, 3.5 mL of a 10% (*w*/*v*) sodium sulfate solution was added, at a speed of 1 mL/min, applying constant sonication (Auto Science, model AS5150B, 180 W, 20 kHz, Lewisville, TX, USA) for 2 h at room temperature. The resulting sample was centrifuged at 3500 rpm for 30 min; the supernatant was discarded. The particles obtained were suspended in distilled water and centrifuged again at 3500 rpm for 30 min. Finally, the particles were suspended in 150 mL of distilled water and kept at room temperature until characterization. 

### 2.3. Characterizations

#### 2.3.1. Scanning Electron Microscopy (SEM)

The microscopic characteristics of the samples were obtained using a JEOL JSM-6610LV (Tokyo, Japan) scanning electron microscope. The samples were gold coated in a Denton Vacuum cathode sputter (Denton Vacuum, Moorestown, NJ, USA), model DEESK V. Prior to observing the ChNPs, a drop of the ChNPs solution was placed on a silica plate and allowed to dry at room temperature. The average size of the nanoparticles was measured using the ImageJ 1.54g software based on the SEM images.

#### 2.3.2. Fourier-Transform Infrared Spectroscopy (FTIR)

The functional groups of chitosan and ChNPs were identified by a Fourier-transform infrared spectrophotometer (Bruker, VERTEX 70, Billerica, MA, USA) with an attenuated total reflectance (ATR) accessory. Approximately 0.05 g of dry sample was placed directly on the ATR glass. The spectrum was obtained in the region of 4000–500 cm^−1^, employing eight scans with a resolution of 4 cm^−1^.

#### 2.3.3. Atomic Force Microscopy, AFM

The size of chitosan nanoparticles was determined using an atomic force microscope (JEOL JSPM-5200, Tokyo, Japan). Before observing the nanoparticles, a drop of the three different ChNPs solutions was placed on a silica plate and allowed to dry at room temperature.

### 2.4. Antifungal Activity

#### 2.4.1. Isolation of the Fungus Colletotrichum Gloeosporioides

The *Colletotrichum gloeosporioides* sample was taken directly from Ataulfo mango fruits (at physiological maturity with symptoms of anthracnose), harvested in an orchard located in the municipality of Técpan de Galeana, in the state of Guerrero, Mexico. The seeding of the diseased tissue was based on the method described by Lemes [26]. Samples were taken from visually damaged tissues, using a previously sterilized razor; 1 cm^2^ fragments of diseased tissue were cut. They were disinfected with a 1% sodium hypochlorite solution for 1 min; the samples were then washed with distilled water and allowed to dry at room temperature on sterile absorbent paper for 3 min. The tissue obtained was placed in Petri dishes with PDA culture medium, using a laminar flow hood. The samples were incubated at 25 ± 3 °C for 5 days, with the boxes previously sealed. The procedure was performed in quintuplicate. The micelles obtained were isolated in fresh PDA to obtain pure cultures.

#### 2.4.2. Characterization of Colletotrichum Gloeosporioides

The incubated samples were analyzed using a Karl Zeiss model Scope A1 optical microscope (Oberkochen, Germany). To do this, 1 mm^2^ of acervula was placed into a slide, to which a drop of lactophenol blue was previously added. The morphology of *Colletotrichum gloeosporioides* was identified by scanning electron microscopy. To achieve that, 2 mm^2^ portions of incubated inoculum were placed in a desiccator for 24 h; they were later coated with gold for analysis.

#### 2.4.3. In Vitro Assays

The preparation of *Colletotrichum gloeosporioides*’ inhibitory solutions was carried out following the method proposed by Liu et al. [20] with some modifications. A total of 5.9 g of potato dextrose agar was added to 150 mL of each solution (0.25%, 0.5% and 1%) of chitosan nanoparticles. Solutions were also prepared by adding 3.9 g of potato dextrose agar to a solution prepared with 100 mL of distilled water and 50 mL of chitosan nanoparticle solution (0.25%, 0.5% and 1%). A solution with 150 mL of distilled water and 5.9 g of potato dextrose agar was used as the control. The seven solutions thus obtained were kept under constant stirring (50 rpm for 10 min); they were subsequently sterilized in a sterilization oven at 121 °C and 15 lb. for 15 min. Finally, they were poured into Petri dishes using a laminar flow hood, where they remained for 48 h. In each of the Petri dishes, a disk of mycelium from the fungus *Colletotrichum gloeosporioides*, 1 cm in diameter, was placed exactly in the center. Incubation was carried out at 25 ± 3 °C for 7 days. Each experiment was performed by triplicate. Potato dextrose agar is described as the most widely used medium in in vitro tests of ChNPs [27].

#### 2.4.4. Fungus Growth

The radial growth of the mycelium was determined with an acetate, from day 3 to day 7, every 24 h; at that time, the control sample showed complete growth. The antifungal index (AI) of the chitosan nanoparticles was determined following the methodology proposed by [28] (Equation (1)).
(1)$$AI\left(\%\right)=\left(1-\left[{DC}_{t}/{DC}_{control}\right]\right)*100.$$
where DC*_t_* = diameter of the colonies in the treated boxes and DC*_control_* = diameter of the colonies in the control boxes. Finally, the speed of pathogen growth was determined using the methodology proposed in [29]; namely, the ratio of relative fungal growth (increase in colony diameter over a time interval) over the elapsed time interval.

### 2.5. Statistic Analysis

A completely randomized design with three repetitions was used. The results of mycelial growth were subjected to ANOVA analysis of variance. Growth velocity was analyzed using simple ANOVA. The means were compared using the Tukey Test (*p* < 0.05); for this, the OriginPro2024b program was used.

## 3. Results and Discussion

### 3.1. Infrared Spectroscopy

Figure 1 contains the infrared spectrum of the chitosan sample and the chitosan nanoparticles. An absorption band between 3500 and 3300 cm^−1^ was observed for the main functional group of chitosan and is due to the stretching vibrations of the O-H group. The presence of absorption bands at 1624 and 1502 cm^−1^ is due to the bending vibration of the N-H group of the protonated amine groups (NH_2_) and the C-H band corresponding to the vibration of the alkyl group. The absorption bands at 1025 and 1062 cm^−1^ were due to the antisymmetric stretching vibration of the COC bridges and the pyranosic ring in the chitosan matrix, respectively. The band of chitosan nanoparticles shifted at 1529 cm^−1^ is due to the movement of NH_2_ bond and the strong band at 1374 cm^−1^ is due to the C–H bending vibration of alkyl group. In the same way, the band corresponding to the pyranosic ring is found at 1042 cm ^−1.^ The ionic interaction with the treated molecules indicates the conversion of the chitosan polymer into the nano form that forms a cross-link with the treated molecules.

### 3.2. Scanning Electron Microscopy

The microscopic characteristics of both the macroscopic chitosan and the ChNPs are shown in Figure 2. The macroscopic chitosan (Ch) has the typical flake morphology, with an average size of 66 µm. However, the structure of Ch changes completely after sonication treatment, resulting in ChNPs that are spherical in shape with a smooth surface and a reduction in particle size (Figure 2b–d); this particle shape is like that obtained using the gelation technique, a more complicated method [30]. Ultrasonication produces a fragmentation effect that causes the particles to rupture due to shock waves generated between the particles from cavitation [31]. The application of ultrasonic radiation results in differing particle sizes, which interact with Tween 20. Other studies report ChNPs with spherical geometry and homogeneous distribution using a solution of tripolyphosphate and chitosan [32,33]. However, the component used (tripolyphosphate) to obtain the chitosan nanoparticles is toxic, in contrast to sonication, which offers a highly efficient approach without significant byproducts.

### 3.3. Atomic Force Microscopy

Figure 3 shows the AFM images of the ChNPs at the three different concentrations. The spherical geometry and its homogeneous distribution can be corroborated. At 0.25%, the average particle size is 58 nm, while at 0.5% the average size is 30 nm; however, at a concentration of 1%, the average size is 70 nm. The ultrasonic preparation of chitosan nanoparticles (present work) yields particles smaller than 100 nm, whereas in other methods the size is more than 200 nm [34]. Studies report that nanoparticles smaller than 70 nm can be absorbed into the nucleus of fungal cells [35]. The results showed that the nanoparticle size and the molecular weight, especially that of the medium-molecular-weight chitosan nanoparticles, was greatly influenced by the concentration of chitosan used in sonication. In another study, ChNPs with an approximate size of 180 nm were obtained from medium-molecular-weight chitosan through ionic gelation, which had a 53% inhibition against *F. graminearum* [36]. Therefore, particle size plays an important role in antifungal properties.

The particle size of the macroscopic chitosan is more dispersed, in the range of 30–110 µm, with an average particle size of 66 µm (Figure 4a). All the ChNPs obtained are smaller than 100 nm in size (Figure 4b); it can be noted that the size distribution is more homogeneous in the smaller particles, which may be due to a greater interaction of the different functional groups of the molecule.

### 3.4. Identification of Colletotrichum Gloeosporioides

The genus *Colletotrichum* has been identified through its morphological characteristics. such as the size and shape of the conidia and the appressorium; the color of the colony, its texture and the growth rate are also used. In the present study, the pathogen *Colletotrichum gloeosporioides* (obtained from Ataulfo mango fruits with anthracnose) was characterized by the formation of colonies whose mycelial growth is radial, white and olive-gray in color, in addition to a salmon-colored conidial mass in the center of the colony (Figure 5a). These results coincide with what was reported in the literature, for this same fungus, but obtained from other sources. This pathogen, isolated from oil palm, has gray colonies with orange conidial masses; its growth is radial in concentric circles [37]; in vitro tests carried out with *Colletotrichum gloeosporioides* isolated from tomato fruits also show conidia with radial growth and a white color, with orange conidial masses [38].

The microscopic morphology of the conidia was observed using an optical microscope (Figure 5b). The characteristic conidia of *Colletotrichum gloeosporioides* were found, which have a cylindrical shape with straight sides and round ends; this is one of the categories in which this genus can be found [39]. This colony was used to carry out antifungal assays using chitosan nanoparticles.

Figure 5c contains an SEM image of the mycelium of *Colletotrichum gloeosporioides* with a healthy appearance, without apparent involvement; the conidial layer is regular and smooth, which is a sign of the softness and integrity of the conidia, as seen in (Figure 5d). The conidia have a cylindrical shape (r = 2 µm, h = 10 µm); these characteristics coincide with those obtained by [40], of the fungus isolated from guava whose conidia length is 8 µm. The elongated hyphae observed in the present work have characteristics like those reported [41] for the pathogen isolated from papaya with anthracnose, whose length is 20 µm.

### 3.5. In Vitro Assays

The inhibition of *Colletotrichum gloeosporioides* by chitosan nanoparticles at different concentrations was recorded; in all cultures with ChNPs, a clear inhibitory effect was observed, compared to the control sample (Figure 6a). This is because the ChNPs are polycationic, with a high surface charge, which allows them to interact more effectively with the fungus. The maximum growth occurred on the seventh day of incubation. At a concentration of 0.5% ChNPs/PDA, the lowest growth of the pathogen was obtained (Figure 6b); on the contrary, with the 1% ChNPs/PDA-H_2_O solution, (Figure 6f) the lowest inhibitory percentage was obtained.

In all cases, a reduction in mycelial growth of the pathogen was observed compared to the control (Figure 5a); the latter reached a radius of 7 cm on the seventh day of incubation. The inhibitory effect of ChNPs, both in PDA and in the solution of PDA-H_2_O, reached a percentage greater than 50. All ChNP solutions prepared in PDA are those with the greatest antifungal effect. From day 5 onwards, the 0.25% and 1% solutions present a similar inhibitory effect; during this time, they only inhibit 78 and 83%, respectively. However, the nanoparticle solution prepared at 0.5% in PDA inhibits 97% of *Colletotrichum gloeosporioides* (Table 1); in this case, the mycelium only grew 0.55 mm, this may be due to their size (30 nm). Due et al. [42] reported that smaller particles exhibit a higher surface area-to-volume ratio, increasing the contact surface between the particle and the fungal membrane. This results in greater bioactivity and the potential for quicker penetration into the cell compared to larger nanoparticles, possibly leading to a more rapid increase in intracellular concentration. This effect is superior to that obtained with ChNPs in combination with 1% (50%) lemon essential oil, applied to *Colletotrichum gloeosporioides* isolated from papaya [43]. It is possible to achieve 82% antifungal action using ChNPs and 0.05% nanche extracts, in *Colletotrichum gloeosporioides* isolated from soursop [44]; however, the effect is smaller than that obtained in the present work using only ChNPs. Finally, the ChNPs in PDA diluted with H_2_O at a concentration of 0.25, 0.5 and 1% showed an inhibition of 84, 86 and 69% on mycelial growth, respectively.

As can be seen in Figure 7, the concentration of 0.5% (30 nm) is the one with the greatest inhibitory effect, both in the solution with PDA and in the solution of PDA with water, these assays had a stronger impact on intracellular processes because they entered cells more easily due to their small size. El-Mohamedya et al. [21] reported similar behavior in chitosan nanoparticles (40 nm) against the fungus *F. solani* isolated from tomato, achieving 100% inhibition at a concentration of 0.5%. Although other concentrations of ChNPs have a lower performance in inhibiting the mycelial growth of *Colletotrichum gloeosporioides*, their inhibitory potential is high.

The growth rate of *Colletotrichum gloeosporioides* where the ChNPs solutions were applied is lower than in the control sample (Table 2). The 0.5% concentration in PDA is the one in which the lowest growth speed was found (0.008 cm/h), 80% lower than in the control sample. It should be noted that this solution is the one with which the smallest micellar radius was obtained (0.4 cm). Like the inhibitory activity, the nanoparticles that are smaller in size (C = 0.5%) affected the germination of spores. This can be explained by the possible agglomeration of particles in the medium, which provides a surface for the spores to adhere to and subsequently germinate. This leads to greater electrostatic interaction between the bioactive compounds and the plasma membrane of the fungi. [45]. The addition of the medium dissolved in water to the ChNPs solution promotes micellar growth (Table 1), which is confirmed by the growth speed (0.01 cm/h) when using 1% ChNPs/PDA-H_2_O. Until now, the growth rate of *Colletotrichum gloeosporioides* where ChNPs are used as an inhibitory agent has not been reported. In Ref. [29], the speed of mycelial growth of chitosan against the fungus *Bipolaris oryzae* was reported, reaching one third of the speed of the control.

All solutions decreased the mycelial growth of *Colletotrichum gloeosporioides* during the 7 days of incubation (Figure 8), inhibiting the germination of the fungus conidia. That is, chitosan nanoparticles have a fungicidal effect on this pathogen, with the concentration of 0.5% being the one with the highest performance. The control sample reached a mycelial growth of 7 cm during the study time. This effect is attributed to the fact that at this concentration, smaller chitosan particles (30 nm) were obtained, which easily penetrate the plasma membrane, reaching the cytosol that contains low levels of Ca^2+^; when these levels interact with chitosan, the homeostatic mechanism is altered, allowing for the free flow of Ca^2+^ gradients, resulting in cellular instability until cell death occurs [46]. From the results obtained, it can be concluded that the concentration of chitosan nanoparticles plays an important role, as suggested by Chowdappa et al. [47], who reported only a 48% inhibition of *Colletotrichum gloeosporioides* at a concentration of 0.5% of nanoparticles of chitosan (DD = 85%, low molecular weight), lower than that obtained in the present work.

Hyphae of *Colletotrichum gloeosporoides* cultivated in media without ChNPs displayed a regular, homogeneous, and smooth surface, branching at a considerable distance from the apical tip of the hyphal cell (Figure 9a). When they were exposed to the various ChNPs, treatments showed significant inhibition of development and pronounced deformation compared to the control. In treatments D, E, and F, distortion, deformity, excessive branching, and the collapse of the hyphae were evident, along with a granular and swollen appearance (Figure 9e–g). With treatments A and C, the hyphae became weak, translucent, and hollow, exhibiting irregular morphology that included thickened sections of reduced size (Figure 9b,d). In contrast, treatment B, which showed the greatest inhibition, resulted in the hyphae breaking and appearing coiled, as well as exhibiting abnormal contortion and branching (Figure 9c). Therefore, the alterations in the morphology of the *Colletotrichum gloeosporoides* pathogen following the different ChNPs treatments are linked to manifestations occurring within the fungal cells, indicating a compromised plasma membrane among a range of other cellular changes that are undoubtedly induced by the ChNPs.

Previous studies have also confirmed alterations to the hyphae of *Colletotrichum* [48], *Rhizopus stolonifera* [49], and *Fusarium oxysporum*, [50] by chitosan nanoparticles. The antifungal mechanism proposed for this study is linked to the fact that ChNPs have a higher affinity to bind to fungal cells due to their high surface charge, interacting more effectively with the fungus compared to the free form of the chitosan polymer, causing a diffusion in the plasma membrane. This alters the morphology of the cell wall and the permeability of the membrane, allowing for the leakage of intracellular material into the environment (Figure 10).

### 3.6. Statistical Analysis

The analysis of compliance with the assumptions of normality and constant variance is shown, using the ANOVA test in mycelial growth. The box plot (Figure 11) allows us to confirm that the highest inhibitory percentage of mycelial growth corresponds to the test carried out with the 0.5% ChNPs/PDA solution. An important inhibitory effect is observed, even from day 3 of inoculation. The analysis of variance gave rise to *p* = 0.000 < 0.05, which allows us to reject the null hypothesis. The previous results also allow us to conclude that all the trials in which ChNPs have been used have great inhibitory potential for *Colletotrichum gloeosporioides*. In the same way, it is evident that all tests with ChNPs were significantly different, compared to the control test, both in PDA and in PDA-H_2_O.

The box plot of the radial growth rate of the *Colletotrichum gloeosporioides* mycelium is shown in Figure 12. As can be seen, test B (0.5% ChNPs with PDA) presents the lowest mycelial growth rate; this result corroborates the percentage of inhibition obtained with this same test (97%). It can also be observed that the median in the control trial is much higher, compared to the trials in which ChNPs were applied; that is, ChNPs present considerable antifungal activity. The radial growth of the mycelium was less than 4 cm in all trials. The analysis of variance gave rise to *p* = 0.000 < 0.05 (Figure 12), which corresponds to the alternative hypothesis. These results were confirmed by the one-factor ANOVA (Tukey test) statistical analysis, where significant reduction in the data compared with the control is observed. The analysis was based on the box plot, with data recorded every 24 h.

Finally, it can be concluded that the different tests at different concentrations of chitosan nanoparticles are significantly different, compared to the control test; that is, any of the concentrations used present good performance to inhibit the mycelial growth of *Colletotrichum gloeosporioides* by up to 97%. The mechanism of action of chitosan nanoparticles can be associated with the molecular weight of the chitosan used (average molecular weight); this molecular weight confers chitosan an important antifungal activity, that is, accelerator activity [51]. The size obtained from the ultrasonic application enhances the inhibitory properties of chitosan; as previously mentioned, smaller sizes facilitate easier penetration into the cells of *Colletotrichum gloeosporioides*, altering their integrity and causing the release of intracellular components. This explains the superior antifungal activity of chitosan nanoparticles compared to their free polymer or solution form. A study conducted by Ma and Lim [52] reported that the cellular uptake of chitosan nanoparticles in the cells was greater than that of chitosan molecules, as the molecules in their bulk form were located extracellularly. This suggests that chitosan nanoparticles could penetrate fungal cells and thus disrupt DNA and RNA synthesis. Minor antifungal action (60%) has been reported when combining chitosan nanoparticles with TEO essential oil, even at higher concentrations (3%) than those used in the present work [53]. Similar effects have been found in *Colletotrichum fragariae* when applying chitosan nanoparticles (26 nm) combined with blueberry extract in PDA, delaying its development [11]. It is possible to achieve 100% inhibition of *Colletotrichum gloeosporioides* and *fragariae* by using high concentrations (>5%) of chitosan particles in thyme essential oil [54].

## 4. Conclusions

In this study, chitosan nanoparticles were obtained using ultrasound and Tween 20 at three different concentrations (0.25, 0.5 and 1% *m*/*v*). The resulting nanoparticles were spherical, with an average particle size of 30 nm (at 0.5%), 58 nm (at 0.25%) and 70 nm (at 1%). A reduction in chitosan particle size was associated with an enhancement in its antifungal properties, as demonstrated by the in vitro test against *Colletotrichum gloeosporioides*. On the seventh day, the control sample was fully colonized by the mycelium, while the 0.5% concentration of ChNPs in PDA exhibited the highest inhibitory effect (97%) compared to the control; this suggests that the vegetative and reproductive structures of the fungus could have undergone irreversible alterations that prevented its survival and adaptation to the environmental conditions. The antifungal activity observed may be linked to the interaction between the ChNPs and the fungal cells, where the nanoparticle surface area facilitates their absorption onto the cell surface, thereby altering the cell’s composition and making essential nutrients unavailable for growth. The relationship between ultrasound application and particle size reduction was confirmed, indicating the potential for property manipulation in future applications. Furthermore, it was found that formulating chitosan in nanoparticle form significantly enhanced its antifungal effectiveness. Consequently, chitosan nanoparticles are anticipated to emerge as a potent and safe natural antifungal agent.

## Figures and Tables

**Figure 1 polymers-16-03058-f001:**
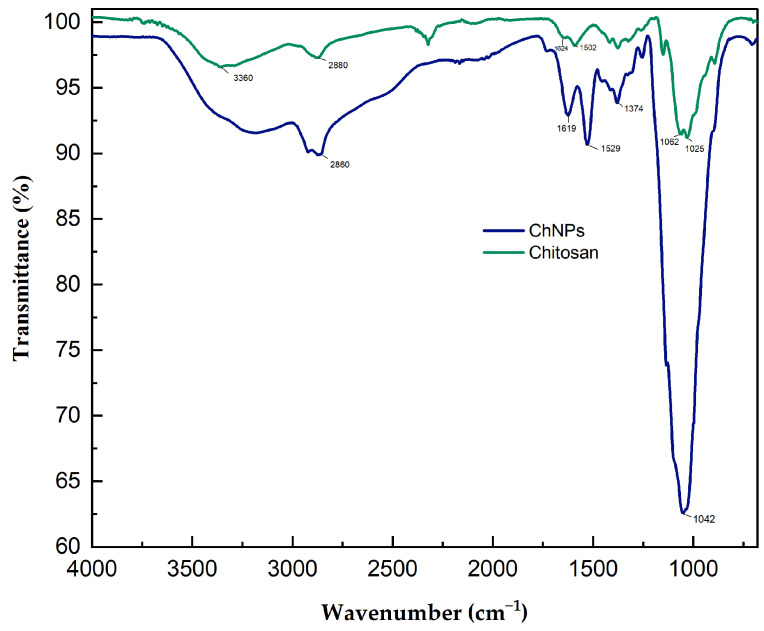
Infrared spectrum of chitosan and chitosan nanoparticles.

**Figure 2 polymers-16-03058-f002:**
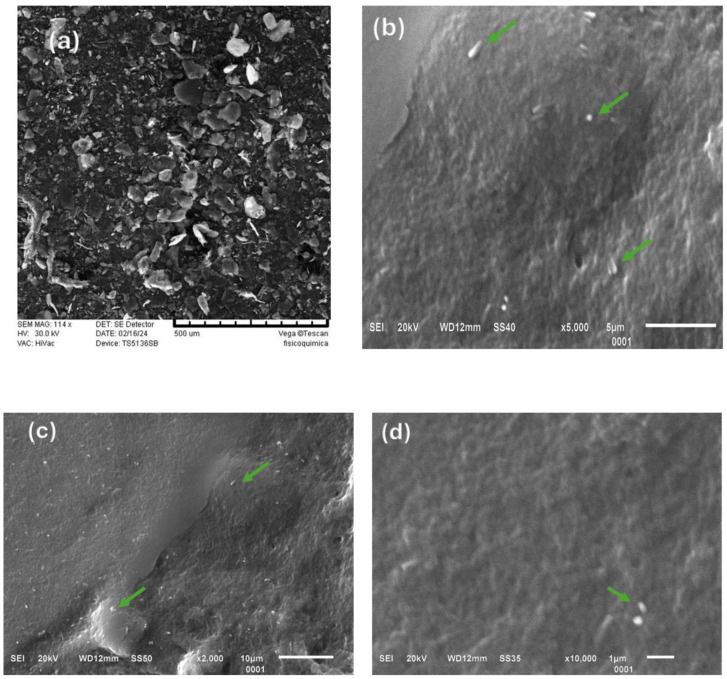
SEM images of macroscopic chitosan (**a**) and ChNPs (**b**–**d**). The arrows indicate the presence of nanoparticles.

**Figure 3 polymers-16-03058-f003:**
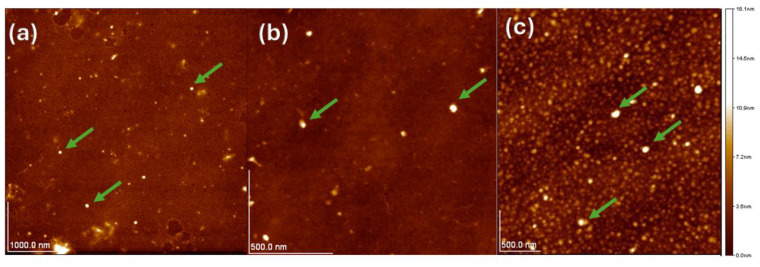
AFM images of ChNPs, at concentration (**a**) 0.25%, (**b**) 0.5%, (**c**) 1%.

**Figure 4 polymers-16-03058-f004:**
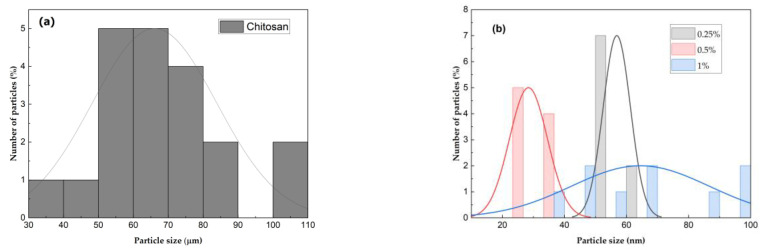
(**a**) Macroscopic particle size distribution of chitosan; (**b**) size distribution of chitosan nanoparticles at the three different concentrations (0.25, 0.5 and 1%).

**Figure 5 polymers-16-03058-f005:**
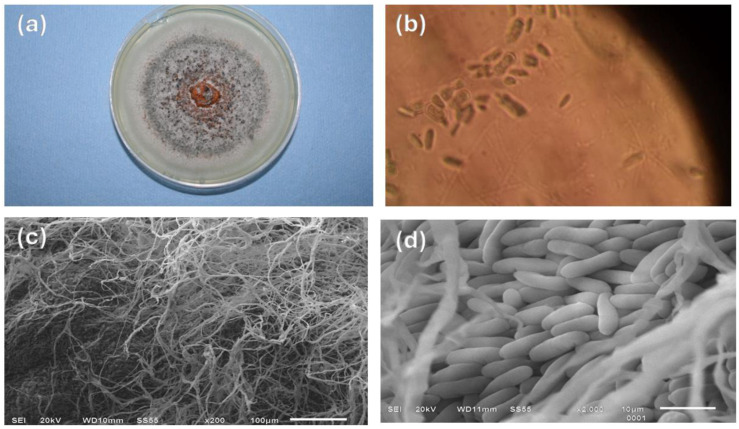
(**a**) *Colletotrichum gloeosporioides* in PDA at 25 °C, (**b**) optical image of the colonial morphology of *Colletotrichum gloeosporioides* (40×), (**c**) SEM of *Colletotrichum gloeosporioides* mycelium, (**d**) SEM micrograph of conidia of *Colletotrichum gloeosporioides*.

**Figure 6 polymers-16-03058-f006:**
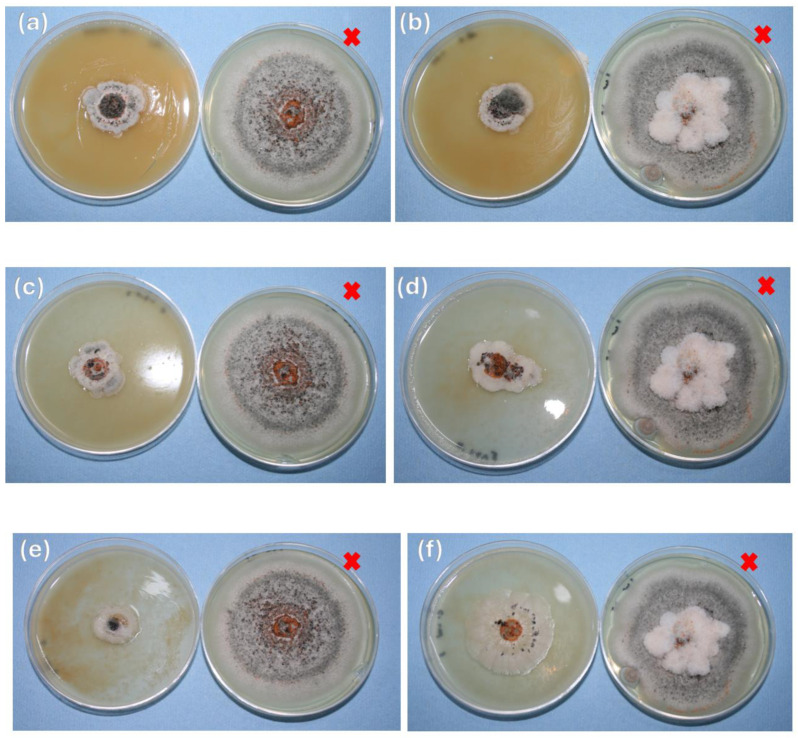
Antifungal activity of ChNPs at different concentrations on the seventh day of incubation. (**a**) 0.25% ChNPs/PDA, (**b**) 0.5% ChNPs/PDA, (**c**) 1% ChNPs/PDA, (**d**) 0.25% ChNPs/PDA-H_2_O, (**e**) 0.5% ChNPs/PDA-H_2_O, (**f**) 1% ChNPs/PDA-H_2_O, versus the control (x) which shows maximum growth.

**Figure 7 polymers-16-03058-f007:**
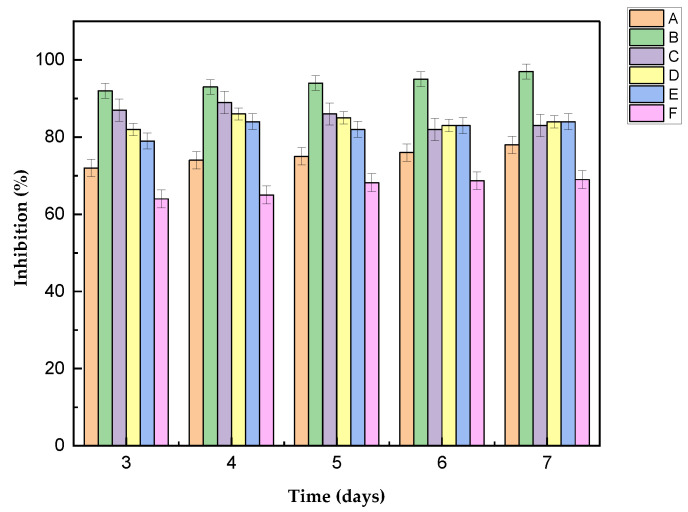
Antifungal activity of chitosan nanoparticles after 7 days of incubation.

**Figure 8 polymers-16-03058-f008:**
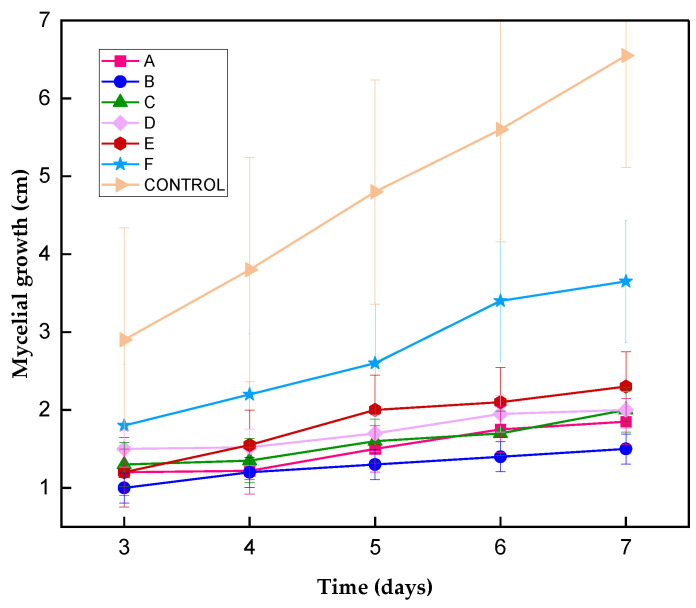
Kinetics of *Colletotrichum gloeosporioides* growth during 7 days of incubation with and without ChNPs.

**Figure 9 polymers-16-03058-f009:**
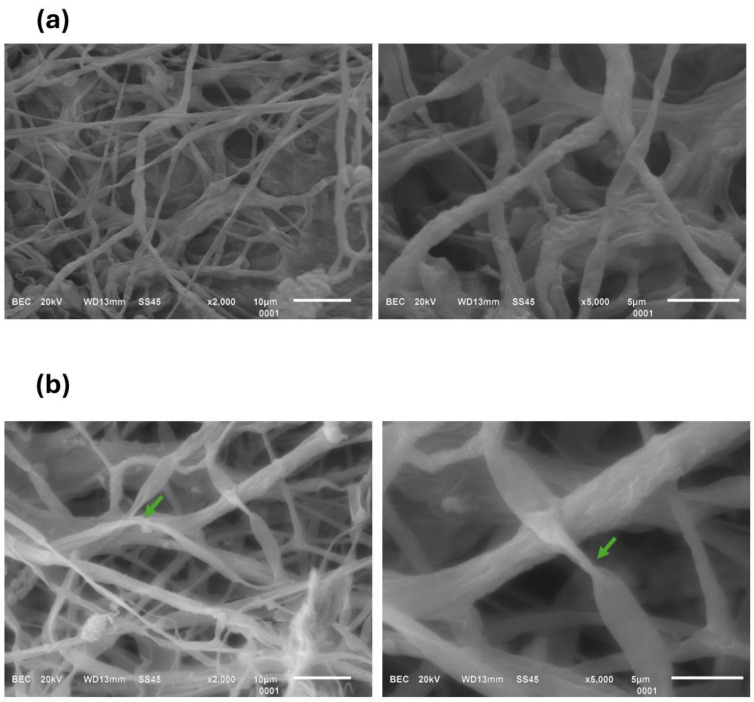
SEM of *Colletotrichum gloeosporioides* after 7 days of incubation with and without ChNPs. (**a**) Control, (**b**) A = 0.25% ChNPs/PDA, (**c**) B = 0.5% ChNPs/PDA, (**d**) C = 1% ChNPs/PDA, (**e**) D = 0.25% ChNPs/PDA-H_2_O, (**f**) E = 0.5% ChNPs/PDA-H_2_O, (**g**) F = 1% ChNPs/PDA-H_2_O. Arrows indicate important changes in the morphology of *Colletotrichum gloeosporioides* after ChNPs treatment.

**Figure 10 polymers-16-03058-f010:**
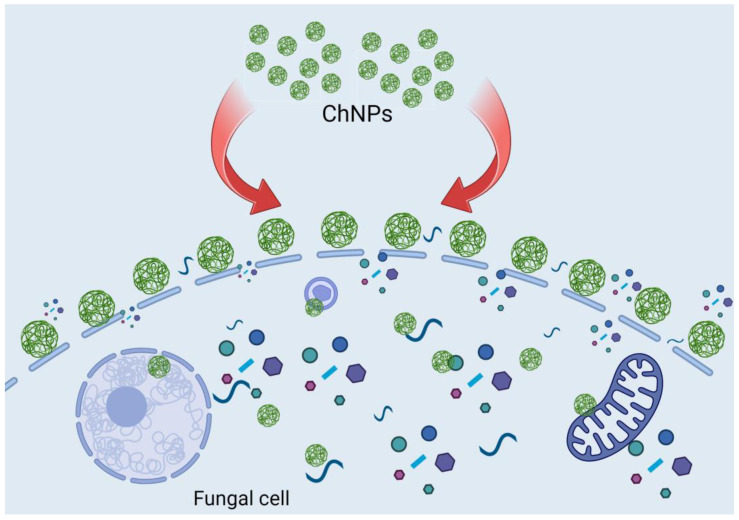
Schematic mechanism of antifungal activity of ChNPs. Chitosan nanoparticles penetrate fungal cell and disturb cell processes.

**Figure 11 polymers-16-03058-f011:**
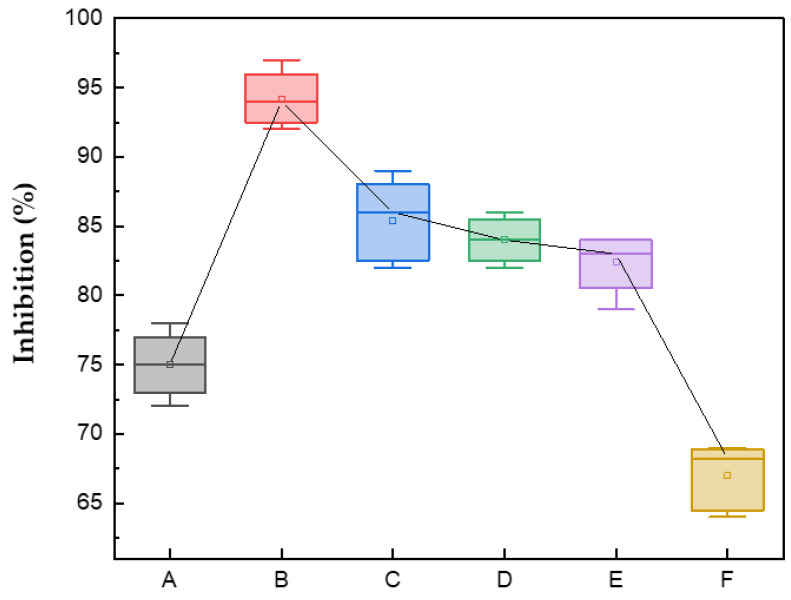
Box plot of the inhibitory effect of ChNPs on the seventh day. ANOVA for speed mycelial growth rate.

**Figure 12 polymers-16-03058-f012:**
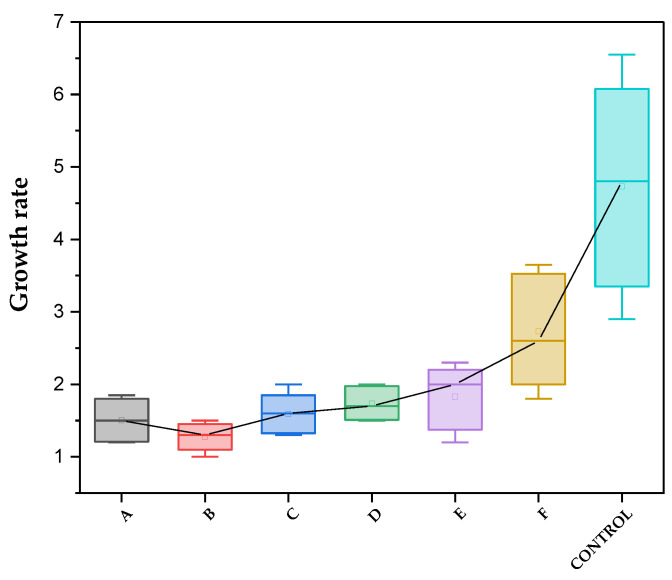
Box plot of ChNPs growth rate on the seventh day of inoculation.

**Table 1 polymers-16-03058-t001:** Inhibition of mycelial growth of *Colletotrichum gloeosporioides* against ChNPs.

Solution	Inhibition of Mycelial Growth by Day (%)				
	3	4	5	6	7
Control	0	0	0	0	0
A 0.25% ChNPS/PDA	72 ^c^	74 ^c^	75 ^c^	76 ^c^	78 ^c^
B 0.5% ChNPS/PDA	92 ª	93 ^a^	94 ^a^	95 ^a^	97 ^a^
C 1% ChNPS/PDA	82 ^b^	83 ^b^	86 ^b^	87 ^b^	89 ^b^
D 0.25% ChNPS/PDA-H_2_O	82 ^b^	86 ^b^	85 ^b^	83 ^b^	84 ^b^
E 0.5% ChNPS/PDA-H_2_O	79 ^b^	81 ^b^	82 ^b^	83 ^b^	86 ^b^
F 1% ChNPS/PDA-H_2_O	64 ^d^	65 ^d^	68.2 ^d^	68.7 ^d^	69 ^d^

Means that do not share a letter are significantly different (*p* < 0.05).

**Table 2 polymers-16-03058-t002:** Growth speed of *Colletotrichum gloeosporioides* against the different ChNPs treatments.

Treatments	Growth Speed of *Colletotrichum gloeosporioides* (cm/h)				
	3	4	5	6	7
Control	0.040 ^a^	0.039 ^a^	0.040 ^a^	0.039 ^a^	0.038 ^a^
A 0.25% ChNPS/PDA	0.016 ^bc^	0.0125 ^bc^	0.0125 ^bc^	0.012 ^bc^	0.01 ^bc^
B 0.5% ChNPS/PDA	0.013 ^c^	0.0125 ^c^	0.01 ^c^	0.009 ^c^	0.008 ^c^
C 1% ChNPS/PDA	0.018 ^bc^	0.014 ^bc^	0.013 ^bc^	0.011 ^bc^	0.012 ^bc^
D 0.25% ChNPS/PDA-H_2_O	0.020 ^bc^	0.015 ^bc^	0.014 ^bc^	0.013 ^bc^	0.012 ^bc^
E 0.5% ChNPS/PDA-H_2_O	0.016 ^bc^	0.016 ^bc^	0.017 ^bc^	0.015 ^bc^	0.014 ^bc^
F 1% ChNPS/PDA-H_2_O	0.026 ^b^	0.023 ^b^	0.021 ^b^	0.02 ^b^	0.019 ^b^

Means that do not share a letter are significantly different (*p* < 0.05).

## Data Availability

The original contributions presented in the study are included in the article, further inquiries can be directed to the corresponding author.

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
