# Peer review of "Ultrasonic Production of Chitosan Nanoparticles and Their Application Against Colletotrichum gloeosporioides Present in the Ataulfo Mango"

_polymers, 2024, doi:10.3390/polym16213058_

Round 1
Reviewer 1 Report (Previous Reviewer 1)
Comments and Suggestions for Authors
The manuscript investigates the ultrasonic synthesis of chitosan nanoparticles (ChNPs) and their antifungal effects against Colletotrichum gloeosporioides, a pathogen affecting Ataulfo mangoes. The study emphasizes the eco-friendly synthesis of ChNPs using ultrasound and Tween 20 and explores their efficacy in inhibiting fungal growth through various concentrations and characterization techniques, including FTIR, SEM, and AFM.
My observations are given below:
The study does not adequately explain the exact mechanism by which Chitosan NPs exert their antifungal effects on fungal cells. A deeper exploration of how the nanoparticles interact at the molecular level with fungal cell walls would strengthen the discussion. Also, The manuscript would benefit from a more comprehensive comparison of the effectiveness of ChNPs synthesized via ultrasound with other existing methods, such as chemical or conventional synthesis techniques. Some experimental conditions, such as the parameters used during the sonication process, are not clearly defined. Providing a detailed methodology would improve reproducibility. The abstract should include specific quantitative results, such as percentage inhibition rates, to provide a concise yet informative summary. The discussion should integrate a more detailed comparison of different synthesis methods and provide a stronger rationale for the observed antifungal activity.
By addressing the above suggestions, the quality and scientific contribution of the work will greatly improve.
Comments on the Quality of English Languagewriting style and grammatical mistakes should be avoided
Author Response
Please, see the attachment

Reviewer 2 Report (New Reviewer)
Comments and Suggestions for Authors
The manuscript polymers-3216927 “Ultrasonic production of chitosan nanoparticles and their application against Colletotrichum gloeosporioides present in the Ataulfo mango” by Ivana Solis Vizcaino et al. describes the prepare chitosan-based nanoparticles with high anti-fungal efficiency by a simple and effective ultrasound-assisted top-down method. The topic of the article is relevant, modern methods of analysis are used in this research. The manuscript may be published after major revision.
Comments and questions:
1) Information on the chemical structure of chitosan and its main characteristics, namely molecular weight and degree of deacetylation, which affect the biological activity, including the role of protonated amino groups in the antifungal and antimicrobial activity of chitosan, should be added to the Introduction.
2) Introduction: The mechanism of antifungal activity should also be described.
3) The molecular weight value of 553.53 kDa is better reduced to 553 kDa.
4) Why was chitosan with a molecular weight of 553,000 chosen and how can molecular weight affect antifungal activity?
5) The size and spherical shape of the chitosan nanoparticles should be confirmed by additional methods such as static or dynamic light scattering and nanoparticle tracking analysis.
6) Why choose the agar diffusion method to determine antifungal activity? This method has some limitations because chitosan nanoparticles are not able to diffuse freely in an agar gel.
7) The term “ultrasound-assisted top-down method” appears once in the manuscript in the Abstract.
Author Response
Please, see the attachment

Reviewer 3 Report (New Reviewer)
Comments and Suggestions for Authors
The application of new approaches to crop protection is definitely a new area that needs to be addressed. The authors have proposed eco-friendly chitosan nanoparticles for the protection of mango fruits against C. gloeosporioides infection. From these studies, it is clear that the proposed chitosan nanoparticle technology is effective in reducing the size of chitosan particles in aqueous medium, which are additionally solubilized with surfactant micelles. The proposed technology is ready for industrial application and can indeed contribute to the protection of mango against fungal infection. The manuscript is interesting, well-written and informative. The authors have carefully followed the reviewers' suggestions and have clearly improved the quality of the manuscript.
Round 2
Reviewer 2 Report (New Reviewer)
Comments and Suggestions for Authors
The authors have responded to all questions and comments. The article may be accepted.
This manuscript is a resubmission of an earlier submission. The following is a list of the peer review reports and author responses from that submission.
Round 1
Reviewer 1 Report
Comments and Suggestions for Authors
The study does not present a significant advancement over existing literature. Similar research, such as the study in Molecules (DOI: 10.3390/molecules27041244), has already been conducted with more comprehensive analysis and broader applications. The manuscript under review does not demonstrate enough novelty or a distinct contribution to justify its publication.
Here are some observations may be considered to improve the manuscript:
1. The introduction lacks a thorough review of recent advancements in chitosan nanoparticle synthesis and their applications in agriculture. Including a more extensive literature review would help in contextualizing the study within the broader field.
2. While the methodology is detailed, some sections, such as the preparation of the ChNPs and the antifungal assays, could benefit from additional clarity. For instance, the exact conditions used for the sonication process (e.g., power, duration) and their impact on particle size and antifungal activity are not fully elaborated.
3. The discussion is somewhat superficial, not fully exploring the implications of the findings or suggesting potential mechanisms of action for the ChNPs. The conclusion could also be more robust, with suggestions for future research directions or practical applications in mango production.
4. The figures and tables are generally clear, but some could benefit from more detailed captions and better integration into the text. For example, figures showing the SEM images of ChNPs should be accompanied by a more detailed discussion in the text regarding the observed morphology and size distribution.
5. The abstract should be revised to better reflect the main findings and implications of the study.
6. Include more references to recent studies on chitosan nanoparticles and their antifungal mechanisms.
7. Discuss the relationship between ChNP concentration and antifungal activity in more detail.
8. Elaborate on the potential mechanisms by which ChNPs exert their antifungal effects, and compare these with other studies in the field.
Comments on the Quality of English Language
In the abstract it is mentioned as "twin20" whereas in body text it is mentioned as "tween20".
Reviewer 2 Report
Comments and Suggestions for Authors
Manuscript does not bring novelty to the field of antimicrobial application of nanochitosan . Physico chemical and biomedical characterization must be significantly improved. Novelty to the field must be clarified instantly in the abstract. I cant recommend manuscript in this form for publication.
Comments on the Quality of English LanguageMinor editing of english is recomended.